# BioLCNet: Reward-modulated Locally Connected Spiking Neural Networks

## Abstract

Recent studies have shown that convolutional neural networks (CNNs) are not the only feasible solution for image classification. Furthermore, weight sharing and backpropagation used in CNNs do not correspond to the mechanisms present in the biological visual system. To propose a more biologically plausible solution, we designed a locally connected spiking neural network (SNN) trained using spike-timing-dependent plasticity (STDP) and its reward-modulated variant (R-STDP) learning rules. The use of spiking neurons and local connections along with reinforcement learning (RL) led us to the nomenclature BioLCNet for our proposed architecture. Our network consists of a rate-coded input layer followed by a locally connected hidden layer and a decoding output layer. A spike population-based voting scheme is adopted for decoding in the output layer. We used the MNIST dataset to obtain image classification accuracy and to assess the robustness of our rewarding system to varying target responses.

## 1 Introduction

For many years, deep convolutional neural network (DCNN) has dominated the field of computer vision and object recognition Goodfellow et al. (2016); LeCun et al. (2015). Although novel methods, such as visual transformers Carion et al. (2020) and very recent MLP-based models Tatsunami & Taki (2021) are threatening its reign, CNN is still the most popular architecture employed for solving visual tasks. However, CNNs lack biological plausibility. First of all, neuron activations in an artificial neural network (ANN) are static real-numbered values, that are modeled by differentiable, non-linear activation functions. This is in contrast to biological neurons that use discrete, and mostly sparse spike trains to transmit information between each other, and in addition to the rate of spikes (spatial encoding), they also use spike timing to encode information temporally Tavanaei et al. (2019). Therefore, a spiking neural network (SNN) is more akin to the neural networks in the brain. Spiking neural networks also require fewer labeled data and operations, which makes them compatible with energy-efficient neuromorphic hardware.

Secondly, the brain is incapable of error backpropagation, as done in traditional ANNs. One issue with error backpropagation in ANNs is the weight transport problem, i.e., the fact that weight connectivity in feedforward and feedback directions is symmetric Liao et al. (2016); Bartunov et al. (2018). Additionally, error feedback propagation that does not affect neural activity is not compliant with the feedback mechanisms that biological neurons use for communication Lillicrap et al. (2020).

Furthermore, although convolutional neural networks has shown great potential in solving any translation-invariant task, its use of weight sharing is biologically problematic. There is no empirical support for explicit weight sharing in the brain Pogodin et al. (2021). However, local connections between neurons is biologically plausible, since neurons in the biological visual system exploit them to have local visual receptive fields Gregor & LeCun (2010). To be compatible with this fact, we also used a locally-connected scheme without explicit weight sharing to design our network. Despite the biological nature of local connections, they mostly underperform convolution-based methods with weight sharing in the visual domain, especially on large-scale datasets Bartunov et al. (2018). This weaker performance may be mainly attributed to the smaller number of parameters and better generalization in CNNs. Fewer parameters in CNNs would also require less memory and computational cost, and would lead to faster training Poggio et al. (2017). Studies are being done to bridge

the performance gap between convolutional and locally-connected networks Lillicrap et al. (2020); Bartunov et al. (2018).

Noting the above considerations, in this paper, we are proposing BioLCNet, a reward-modulated locally-connected spiking neural network. Our network is trained using the unsupervised spike-timing-dependent plasticity and its semi-supervised variant reward-modulated STDP. The input images are encoded proportional to the pixels intensity using Poisson rate-coding that converts intensity to average neuron firing rate in Hertz. In the output layer, there are neuronal groups for each class label, and decision making is based on aggregated number of spikes during the decision period. Our novel *dynamic reward prediction error (RPE)* mechanism exploits strongly supported empirical findings to improve classification performance. We test the classification capabilities of our network with different sets of hyperparameters on the MNIST dataset LeCun et al. (1999). We also conduct a classical conditioning experiment to prove the effectiveness of our decoding scheme and rewarding mechanisms.

## 2 RELATED WORK

Neuroscientists and deep learning researchers have long been searching for more biologically plausible deep learning approaches in terms of neuronal characteristics, learning rules, and connection types. Regarding neuronal characteristics, researchers have turned to biological neuronal models and spiking neural networks. The vanishing performance gap between deep neural netwroks (DNNs) and SNNs, and the compatibility of SNNs with neuromorphic hardware and online on-chip training Schemmel et al. (2010) has piqued the interest of researchers Mozafari et al. (2019). For comprehensive reviews on deep learning in spiking neural networks, see Tavanaei et al. (2019); Pfeiffer & Pfeil (2018).

Spiking neurons are activated by discrete input spike trains. This differs from artificial neurons used in an ANN that have differentiable activation functions and can easily employ backpropagation and gradient-based optimization. There are works that use gradient-based methods with SNNs Kheradpisheh & Masquelier (2020); Wu et al. (2018); Neftci et al. (2019); Bellec et al. (2020) and some of them have achieved great performances. On the other hand, many works in this area use derivations of the Hebbian learning rule where changes in connection weights depend on the activities of the pre and post-synaptic neurons Hebb (1949). Spike-timing-dependent plasticity (STDP) and its variants, apply asymmetric weight updates based on the temporal activities of neurons. Normal STDP requires an external read-out for classification Mozafari et al. (2018), and have been applied to image reconstruction and classification tasks by many researchers. Some have employed fully-connected architectures Beyeler et al. (2013); Tavanaei & Maida (2015); Allred & Roy (2016), while others used convolutional layers for feature extraction Masquelier & Thorpe (2007); Panda & Roy (2016); Kheradpisheh et al. (2016; 2018). Reward-modulated STDP (R-STDP) uses a reward (or punishment) signal to directly modulate the STDP weight change, and can be used to decode the output without an external cue. Izhikevich (2007) solved the distal reward problem in reinforcement learning by using a version of R-STDP with decaying eligibility traces that gives recent spiking activity more importance. Around the same time, Florian (2007) showed that R-STDP can be employed to solve a simple XOR task with both rate and temporal encoding of the output. Also, Caporale & Dan (2008) used R-STDP to generate specific spiking patterns in the output of their spiking network. Historically, R-STDP was first adopted with temporal (rank-order) encoding for image classification Mozafari et al. (2018). They employed a convolutional architecture based on Masquelier & Thorpe (2007) and a time-to-first-spike decoding scheme. An extended architecture was later developed which had multiple hidden layers Mozafari et al. (2019). The use of R-STDP with Poisson rate-coding has been mostly limited to fully-connected architectures for solving reinforcement learning robot navigation tasks Shim & Li (2017); Bing et al. (2019). To our knowledge, image recognition problems have not yet been addressed by combining R-STDP and rate-based encoding.

The most prevalent architectures used for image classification in deep learning with both DNNs and SNNs are based on convolutional layers and weight sharing. However, there are arguments against the biological plausibility of these approaches Bartunov et al. (2018); Pogodin et al. (2021). Locally connected (LC) networks are an alternative to the convolutional ones. Illing et al. (2019) show that shallow networks with localized connectivity and receptive fields perform much better than fully-connected networks on the MNIST benchmark. However, Bartunov et al. (2018) showed

that the lower generalization of LC networks compared to CNNs results in their underperforming CNNs in most image classification tasks, and prevents their scalability to larger datasets such as ImageNet Deng et al. (2009). Very recently, Pogodin et al. (2021) proposed bio-inspired dynamic weight sharing and adding lateral connections to locally-connected layers to achieve the same regularization goals of weight sharing and normal convolutional filters. The first work to integrate a locally-connected (LC) layer into an SNN Saunders et al. (2019) used a network with no hidden layers where the rate-coded input is passed to the output layer via local connections. They exploited recurrent inhibitory connections similar to the ones employed by Diehl & Cook (2015) to simulate a winner-take-all (WTA) inhibition mechanism in their output. Their learning rule is STDP, and therefore an external readout, in this case n-gram voting, is required for classification. Their network scheme was inspiring in designing our locally connected hidden layer.

## 3 THEORY

In this section, we will outline the theoretical foundations underlying our proposed method. Specifically, the dynamics of the spiking neuronal model, the learning rules used, and the connection type employed in our network will be described.

### 3.1 ADAPTIVE LIF NEURON MODEL

The famous leaky and integrate fire neuronal model is governed by the following differential equation Gerstner et al. (2014),

$$\tau_m \frac{du}{dt} = -[u(t) - u_{rest}] + RI(t), \tag{1}$$

where $u(t)$ denotes the neuron membrane potential and is a function of time, $R$ is the membrane resistance, $I(t)$ is any arbitrary input current, and $\tau_m$ is the membrane time constant. Equation (1) dictates that the neuron potential exponentially decays to a constant value $u_{rest}$ over time. When a pre-synaptic neuron fires (spikes), it generates a current that reaches its post-synaptic neurons. In the simple leaky integrate and fire (LIF) model, a neuron fires when its potential surpasses a **constant** threshold $u_{thr}$. After firing, the neuron's potential resets to a constant $u_{reset}$ and will not be affected by any input current for a period of time known as the refractory period ($\Delta t_{ref}$).

A variant of the LIF model uses adaptive firing thresholds. In this model, $u_{thr}$ can change over time based on the neuron's rate of activity Diehl & Cook (2015). When a neuron fires, its tolerance to the input stimuli and consequently its firing threshold increases by a constant amount, $g_0$, otherwise the threshold decays exponentially with a time constant $\tau_g$ to the default threshold $u_{thr_0}$. Equations (2) to (4) explain the dynamics of the adaptive LIF model,

$$u_{thr}(t) = u_{thr_0} + g(t), \tag{2}$$

where,

$$\tau_g dg/d_t = -g(t), \tag{3}$$

and

$$spike \Rightarrow g(t) = g(t-1) + g_0, \tag{4}$$

### 3.2 REWARD-MODULATED STDP

Spike-timing-dependent plasticity is a type of biological Hebbian learning rule that is also aligned with human intuition ("Neurons that fire together wire together." (Lowel & Singer, 1992)). The normal STDP is characterized by two asymmetric update rules. The synaptic weights are updated based on the temporal activities of pre and post-synaptic neurons. When a pre-synaptic neuron fires shortly **before** its post-synaptic neuron, the causal connection between the first and the second neuron temporal activity is acknowledged, and the connection weight is increased. On the other hand, if the post-synaptic neuron fires shortly **after** the pre-synaptic neuron, the causality is undermined and the synaptic strength will decrease Hebb (1949). These weight updates, called long-term potentiation (LTP) and long-term depression (LTD), can be performed with asymmetric learning rates to adapt the learning rule to the excitatory to inhibitory neuron ratio or the connection patterns of a specific neural network. A popular variant of STDP that integrates reinforcement learning into

the learning mechanism of spiking neural networks is reward-modulated STDP (also known as R-STDP or MSTDP Florian (2007)). In R-STDP, a global reward or punishment signal, which can be a function of time, is generated as the result of the network's activity or task performance. Using a notation similar to Florian (2007), to mathematically formulate both STDP and R-STDP, we can define the spike train of a pre-synaptic neuron as the sum of Dirac functions over the spikes of the post-synaptic neurons,

$$\Phi(t) = \sum_{\mathscr{F}_i} \delta(t - t_i^f). \tag{5}$$

where $t_i^f$ is the firing time of the $i^{th}$ post-syanptic neuron. Now, we can define the variables $P_{ij}^+$ and $P_{ij}^-$ to respectively track the influence of pre or post-synaptic spikes on weight updates. Now, the spike trace $\xi$ for a given spike from neuron $i$ to $j$ can be defined as below,

$$\xi_{ij} = P_{ij}^+ \Phi_i(t) + P_{ij}^- \Phi_j(t), \tag{6}$$

where: (assuming the same ,

$$dP_j^+/dt = -P_j^+/\tau_+ + \eta_{post}\Phi_j(t), \tag{7}$$

$$dP_i^-/dt = -P_i^-/\tau_- - \eta_{pre}\Phi_i(t), \tag{8}$$

where we assumed that $P_{ij} = P_j$ for all pre-synaptic connections related to neuron $j$, and $P_{ij} = P_i$ for all post-synaptic connections related to neuron $i$.

The variables $\tau_\pm$ are the time constants determining the time window in which a spike can affect the weight updates. Using larger time constants will cause spikes that are further apart to also trigger weight updates. The variables $\eta_{post}$ and $\eta_{pre}$ determine the learning rate for LTP and LTD updates respectively. We denote the reward or punishment signal with $r(t)$. The R-STDP update rules for positive and negative rewards can be written as,

$$\frac{dw_{ij}(t)}{dt} = \gamma r(t)\xi_{ij}(t), \tag{9}$$

where $\gamma$ is a scaling factor. The update rule for normal STDP can also be written as,

$$\frac{dw_{ij}(t)}{dt} = \gamma \xi_{ij}(t). \tag{10}$$

Based on Equation (9), we note that R-STDP updates only take effect when a non-zero modulation signal is received at time step $t$. However, STDP updates do not depend on the modulation signal, and are applied at every time step. In other words, STDP can be considered a special case of R-STDP where the reward function is equal to 1 in every time step. This causes STDP to respond to the most frequent patterns regardless of their desirability.

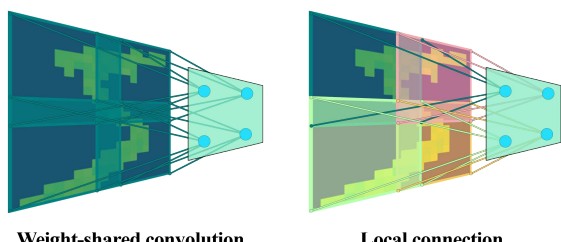

**Weight-shared convolution**          **Local connection**

Figure 1: Visual comparison of convolutional and local connections for a given filter; in convolutional connections, the weights are shared between all receptive fields. However, in a local connections, each receptive field has its own set of weights.

### 3.3 LOCAL CONNECTIONS

A local connection in a neural network is similar to a convolutional connection but with distinct filters for each receptive field. As seen in Fig. 1, in normal convolutional connections, there is one

filter for each channel that is convolved with all receptive fields as it moves along the layer's input. This filter has one set of weights that are updated using the network's update rule. However, In local connection (LC), after taking each stride, a new set of parameters characterize a whole new filter for the next receptive field. This type of connectivity between the input and the LC layer resembles the physical structure of retinal Ganglion cells. Because there are more filters in an LC, the number of distinct synapses in a local connection is greater than a convolutional connection, yet much lower than a dense connection. Similar to a convolutional connection, assuming square filters, and equal horizontal and vertical strides, we can specify a local connection by the number of channels (filters) ($ch_{lc}$), the kernel size ($k$), and the stride ($s$).

## 4 ARCHITECTURE AND METHODS

BioLCNet consists of an input layer, a locally connected hidden layer, and a decoding layer. Each layer structure and its properties alongside the training and rewarding procedure will be delineated in this section. A graphical representation of our network is presented in Fig. 2. The simulation time $T$ is divided into three phases, adaptation period ($T_{adapt}$), decision period ($T_{dec}$), and learning period ($T_{learn}$). The details of each phase will be specified in the remainder of this section.

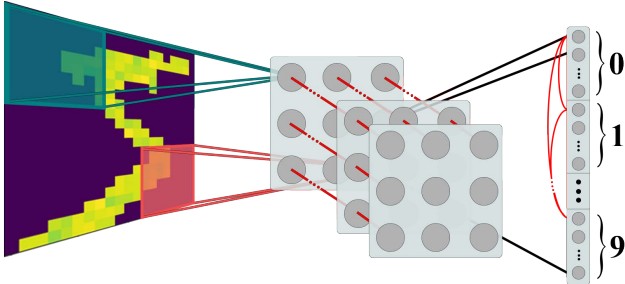

Figure 2: Graphical representation of the proposed network; locally connected filters will be applied to the rate-coded input image. Based on a winner-take-all inhibition mechanism, the most relevant features from each receptive field transmit their spikes to the decoding layer, which selects the most active neuronal group as the predicted label exploiting lateral inhibitory connections. The red lines indicate inhibitory connections.

### 4.1 ENCODING LAYER

The input of the network is an image of dimensions ($ch_{in}, h_{in}, w_{in}$). For a grayscale image dataset such as MNIST, $ch_{in}$ equals to one. Each input channel is rate-coded using a Poisson encoding scheme, i.e, the spiking neuron corresponding to each pixel has an average firing rate proportional to the intensity of that pixel. By choosing the maximum firing rate $f_{max}$, the spike trains average firing rates will be distributed in the interval $[0, f_{max}]$ Hertz based on the pixel values.

### 4.2 FEATURE EXTRACTION LAYER (LOCAL CONNECTIONS)

The encoded input at each simulation time step passes through local connections with $ch_{out}$ distinct filters for each receptive field. Therefore, the output of this layer will have dimensions ($ch_{out}, h_{out}, w_{out}$), where the output size depends on the size of the kernel and the stride. There are generally two approaches in the SNN literature for training a feature extraction layer with rate-coded inputs using STDP to attain a rich feature representation and also prevent the weights from growing too large. One is allowing the weights to have negative values, which corresponds to having inhibitory neurons, as done in the convolutional layers used by Lee et al. (2018). The other is to use a combination of recurrent inhibitory connections and adaptive thresholds as done by Diehl & Cook (2015); Saunders et al. (2018; 2019). In this work, we used the latter approach for our feature extraction LC layer. We use adaptive LIF neurons and inhibitory connections between neurons that share the same receptive

field. This is equivalent to the winner-take-all inhibition mechanism which causes a competition between neurons to select the most relevant features. The inhibitory connections are non-plastic and they all have a static negative weight $w_{inh}$ with a large absolute value.

In normal STDP, the LTP learning rate ($\eta_{post}$) is usually chosen larger than the LTD rate ($\eta_{pre}$) to suppress the random firing of neurons that triggers many LTD updates during the early stages of training. However, this may become problematic in the later stages, and the weights may grow too large. Therefore, in practice, different mechanisms, such as weight clipping and normalization are used to prevent the weights running amok. In this work, we clipped the weights to stay in the range $[0, 1]$. We also employed the normalization technique used by Saunders et al. (2019) and normalized the pre-synaptic weights of each neuron in the LC layer to have a constant mean of $c_{norm}$ at the end of each time step.

### 4.3 DECODING LAYER AND REWARDING MECHANISMS

The final layer of our network is a fully connected layer for reward-based decoding. The layer is divided into $n_c$ neuronal groups where $n_c$ is the number of classes related to the task. Consequently, the $n_{out}$ neurons in this layer are divided equally into $n_c$ neuronal groups. The predicted label for a given test sample is the class whose group has the most number of spikes aggregated over the decision period ($T_{dec}$). This decoding layer is trained using reinforcement learning and R-STDP during the learning period ($T_{learn}$) based on the modulation signal generated by the rewarding mechanism. We designed two different rewarding mechanisms, **static** and **dynamic reward prediction error (RPE)**. In the *static* mechanism, we use a fixed reward or punishment signal for the whole learning period ($T_{learn}$) based on the prediction of the network for the $i^{th}$ training sample,

$$r_i = \left\{ \begin{array}{rc} 1 : & predicted\ label = target\ label \\ -1 : & otherwise \end{array} \right. \tag{11}$$

The second mechanism, *dynamic RPE* is based on the reward prediction error theory in reinforcement learning. According to this theory, the dopaminergic neurons in the brain release dopamine proportional to the difference between the actual reward and the expected reward (not solely based on the actual reward) Schultz et al. (1997); Sutton & Barto (2018). We formulate our *dynamic RPE* mechanism as below,

$$R_i = R_{i-1} - \eta_{rpe}(r_i - \text{EMA}_R) \tag{12}$$

where $R_i$ is the scalar R-STDP modulation signal used during the whole learning period ($T_{learn}$) of the $i^{th}$ training sample, $r_i$ is the reward signal received based on the prediction, and $\text{EMA}_R$ is the exponential moving average of the modulation signals with a smoothing factor $\alpha$.

### 4.4 TRAINING PROCEDURE

The network is trained in a layer-wise fashion. After initializing the weights uniformly between $[0, 1]$, we train the feature extraction LC layer in a completely unsupervised manner using STDP. Simulation time for training the feature extraction layer is $T_{learn}$ time steps. After this layer is trained, the weights are frozen, and we train the decoding FC layer in a semi-supervised manner using R-STDP and the selected rewarding mechanism. Training this layer requires all three simulation phases. The input image is first presented to the network for $T_{adapt}$ time steps to let the LC layer neurons adapt to the input image and select its relevant features. During $T_{dec}$ time steps, the decoding layer accumulates the number of spikes received by each neuronal group to determine the predicted label. Afterwards, the modulation signal is generated and the decoding layer weights are updated using R-STDP for $T_{learn}$ time steps.

When training the LC layer, we observed that after a specific number of iterations (training samples), the weights of this layer converge and remain constant. Fig. 3a visualizes the filters learned after 2000 iterations for 100 filters of size 15 with a stride of 4 applied to the input images. This fast convergence is an evidence showing the strength of STDP learning. Considering these observations, and to save computation time, we limit the number of training sample of the LC layer to 2000 for all of the hyperparameter configurations. Given an input image (Fig. 3b), we can plot the activation map of the LC layer (Fig. 3c). This map shows the post-synaptic neurons corresponding to the relevant features activate, and suppress the other neurons in accordance with the WTA inhibition mechanism.

The network is implemented using PyTorch Paszke et al. (2019), and mostly on top of the BindsNet framework Hazan et al. (2018) to make our code more efficient. We reimplemented the local connection topology to make it compatible with multi-channel inputs and a possible deep extension of our network.

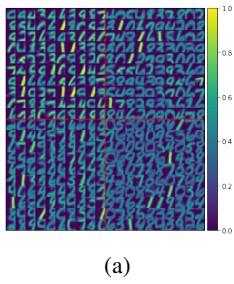 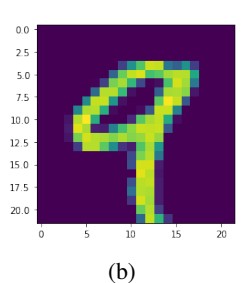 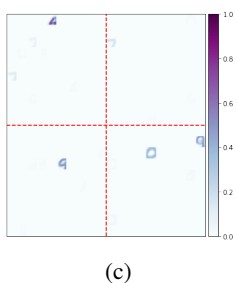

(a) (b) (c)

Figure 3: Input and LC layer visualizations. (a) LC layer learned filters; the red lines separate filters corresponding to each receptive field. (b) A sample input image. (c) The LC layer activation map corresponding to the sample input image shown.

Table 1: BioLCNet (hyper-)parameters; best-performing value for (hyper-)parameters subject to grid search are in bold.

| Parameter | Value |
|---|---|
| $u_{thr_0}$ | -52 $(mV)$ |
| $u_{rest}, u_{reset}$ | -65 $(mV)$ |
| $g_0$ | 0.05 $(mV)$ |
| $\tau_g$ | $10^6$ $(ms)$ |
| $\Delta t_{ref}$ | 5 $(ms)$ |
| $\tau_m$ | 20 $(ms)$ |
| $f_{max}$ | 128 $(Hz)$ |
| $h_{in}, w_{in}$ | 22 |
| $n_{out}$ | [100, 500, **1000**] |
| $ch_{lc}$ | [25, 50, **100**, 250] |
| $k$ | [11, 13, **15**, 17] |
| $s$ | [2, 3, **4**] |
| $T_{adapt}, T_{dec}, T_{learn}$ | 256 $(ms)$ |
| $(\eta_{pre}, \eta_{post})_{STDP}$ | (0.0001, 0.01) |
| $(\eta_{pre}, \eta_{post})_{R-STDP}$ | (0.1, 0.1) |
| $\gamma$ | 1 |
| $\eta_{rpe}$ | [(static), 0.075, **0.125**, 0.175, 0.25] |
| $\alpha$ | 0.9 |
| $w_{inh}$ | -100 |
| $c_{norm}$ | 0.25 |

## 5 EXPERIMENTS AND DISCUSSION

### 5.1 IMAGE CLASSIFICATION

To evaluate our network's classification performance, we trained our model on the MNIST benchmark. Some of the hyperparameters were fixed and others were subject to grid search. The full list of hyperparameters are given in Table 1.

Considering the hyperparameters mentioned in Table 1, we report in Table 2, the classification accuracy on the whole MNIST test set (10000 samples) for four hyperparameter configurations chosen based on the highest test accuracy obtained after conducting a grid search. The number of neurons and synapses for each model are also reported in this table. The final models were all trained using 10000 training samples from the MNIST training set. Using more training samples did not improve the classification performance as can be observed from Fig. 4. The mean and standard deviations

Table 2: MNIST test dataset accuracies obtained by four different sets of hyper-parameters; the test accuracies are averaged over ten independent runs

| Parameters $[k, s, \eta_{rpe}, n_{out}]$ | $n_{neurons}$ | $n_{synapses}$ | Test accuracy | SVM test accuracy |
|---|---|---|---|---|
| [13, 3, 0.025, 100] | 1700 | 430400 | 61.30 ±3.14 | 87.5±1.32 |
| [15, 4, 0.175, 1000] | 1884 | 490000 | 75.00 ±2.68 | 83.3±1.74 |
| [15, 4, 0.125, 1000] | 1884 | 490000 | 76.40 ±2.43 | 83.3±1.74 |
| [15, 4, (static), 100] | 984 | 130000 | 68.8 ±2.87 | 83.3±1.74 |

Table 3: MNIST test dataset accuracies obtained by different SNN approaches

| Paper | Encoding | Architecture | Bio-plausibility criteria | Acc. |
|---|---|---|---|---|
| BioLCNet (proposed, RL) | rate-based | Locally connected+Dense | STDP, RL, LC | 76.40 |
| BioLCNet (proposed, SVM) | rate-based | Locally connected | STDP, LC | 87.5 |
| Beyeler et al. (2013) | rate-based | Dense | STDP | 91.60 |
| Diehl & Cook (2015) | rate-based | Dense | STDP | 95.00 |
| Tavanaei & Maida (2015) | rate-based | Dense | STDP | 75.93 |
| Allred & Roy (2016) | rate-based | Dense | STDP | 86.59 |
| Kheradpisheh et al. (2018) | rank-order | Convolutional | STDP | 98.40 |
| Saunders et al. (2018) | rate-based | Convolutional | STDP | 84.23 |
| Lee et al. (2018) | rate-based | Convolutional | STDP | 91.1 |
| Mozafari et al. (2019) | rank-order | Convolutional | STDP, RL | 97.2 |
| Saunders et al. (2019) | rate-based | Locally connected | STDP, LC | 95.07 |

reported are estimated from ten independent runs. In addition to the RL-based models, another classification approach was employed. In this approach, for each training sample, we create a feature vector containing the number of spikes aggregated over $T_{learn}$ time steps for every filter in the LC layer. We use these feature vectors to train a support vector machine (SVM) classifier. The SVM results are also obtained by training on 10000 training samples, and testing on the whole MNIST test set. The SVM test results for two different hyperparameter configurations are reported in Table 2 and are compared to the RL-based results. The best performance of SVM and RL-based classification are 87.50, and 76.40 respectively. Table3 compares the MNIST test performance obtained by different SNN approaches along with the bio-plausibility criteria to which they adhere.

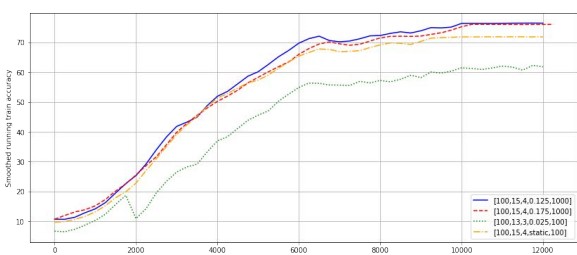

Figure 4: Smoothed running accuracy over the training set for four sets of hyperparameters using the R-STDP classifier

Overall, the supervised SVM has achieved a better performance than the R-STDP method. Two important observations can be made from Table 2. First, the classification accuracy has a positive correlation with the filter size, and the number of neurons in the decoding layer. Secondly, the *dynamic RPE* mechanism improved the classification performance compared to the default *static* rewarding mechanism. *dynamic RPE* plays a similar role to the adaptive learning rate method employed by Mozafari et al. (2018), yet with more biological roots and empirical support.

## 5.2 Classical conditioning

In order to show the effectiveness of our rewarding mechanism, we perform a classical (Pavlovian) conditioning experiment. This type of conditioning pairs up a neutral stimulus with an automatic

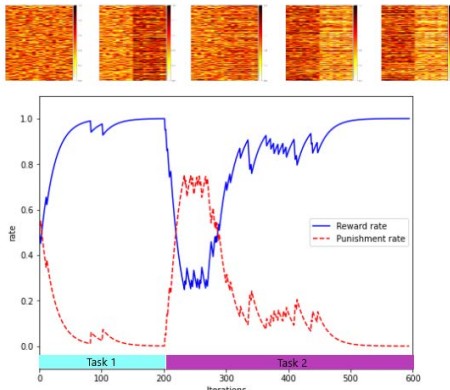

Figure 5: Classical conditioning experiment; in this experiment, we tested the adaptability of the network to varying target responses. The plot shows the rate of receiving reward and punishment averaged over 20 runs, and the decoding layer weight maps at iterations 0, 200, 300, 400, and 600. The right side of the weight maps correspond to the task 1 target response neurons, and the left side corresponds to the task 2 target response neurons. The weights adapt to the varying target response during the experiment.

conditioned response by the agent. In this experiment, we present the network with images belonging to one class of the MNIST dataset as the neutral stimuli. We used the pre-trained feature extraction layer of the network with 25 filters of size 13 and stride of 3, following by a decoding layer with 20 neurons for a two-class prediction task. In the first half of the experiment (task 1), the target response is class 1, and the network receives a constant reward of 1 if it predicts this class regardless of the input. A punishment signal of -1 is received if the agent predicts class 0. We monitor the rate of the reward and punishment received during the experiment. After the convergence in about 50 iterations, Fig. 5 shows that the agent has become completely conditioned on the rewarding response. After 200 iterations, we swap the rewarding and punishing classes, and continue running the network. In task 2, the network should predict the input images as class 0. The RL agent (the network) adapts to the change notably fast, and completely changes its behavior after about 100 iterations. The heat maps in Fig. 5 visualize the weights of the output layer through the training.

The reward adaptability of an RL agent is critical because in many real-world problems the environment is non-stationary. Integration of reward adaptation into spiking neural networks, as done in this work, can pave the path for models that simulate human behaviour with the same spike-based computation as done in the human brain.

## 6 CONCLUSIONS AND FUTURE WORK

In this work, we examined the capabilities of a neural network with three-fold biological plausibility; spiking neurons, local visual receptive fields, and a reward-modulated learning rule. The R-STDP learning rule has been only used for sequential decision making or temporal-coded visual tasks. As the first work to employ R-STDP in locally connected SNNs, we did not expect to achieve state-of-the-art performance. However, we hope that using the novel dynamic RPE rewarding mechanism alongside the emerging local connection scheme will make the future prospects of biological learning rules and architectures in solving real-world problems, more promising.

In the future, by bringing ideas such as dynamic weight sharing and lateral connections Pogodin et al. (2021) to spiking neural networks, we may be able to obtain richer feature representations using locally connected SNNs. We can also exploit the recent advances in SNN minibatch processing Saunders et al. (2020) and neuromorphic hardware Schemmel et al. (2010) to extend our network with deeper architectures and solve more complex tasks.

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
