# OpenReview forum: "BioLCNet: Reward-modulated Locally Connected Spiking Neural Networks"
_ICLR.cc/2022/Conference — ICLR 2022 Submitted_

### Official Review · Reviewer_b4gF · 2021-11-01

**Correctness:** 3
**Technical Novelty And Significance:** 3
**Empirical Novelty And Significance:** 2
**Recommendation:** 6
**Confidence:** 3

**Main Review:**

## Strengths
- The paper is adequately written, and the research is motivated and
  presented clearly.
- To the best of my knowledge, the proposed architecture is a novel
  combination of existing approaches, and the reported results
  convincingly show that it achieves reasonable performance on the
  benchmark task.
- The problem of devising flexible architectures that can perform
  local (Hebbian) learning is one of significant theoretical and
  practical interest, especially with an eye to potential neuromorphic
  applications.

## Weaknesses
- The paper seems rather incremental, in the sense that its main
  innovations seem to be tweaks or variations of existing
  approaches. The Introduction section ends with this sentence: *"The
  proposed network is the first to employ reinforcement learning with
  Poisson rate-coded inputs for image recognition and the first
  locally-connected SNN with a hidden layer."* Even if taken at face
  value, being the first paper to "employ reinforcement learning with
  poisson rate-coded inputs for image recognition" does not seem such
  an impressive primate, if it means that there already exist papers
  where reinforcement learning with poisson rate-coded inputs is
  applied to a different domain, or where rate-coded inputs for image
  recognition are used with a different learning scheme. Moreover, a
  cursory search brought up https://doi.org/10.3389/fncom.2021.543872,
  where reinforcement learning and Poisson rate-coded inputs are
  applied to an MNIST classification benchmark. I haven't searched
  more, but this should suffice to show that the claim may not even be
  true (I'd like for the authors to comment on this). As for being the
  first locally-connected SNN with a hidden layer (again assuming that
  the claim is true), I'm similarly unsure how significant an
  achievement that is, if there already exist convolutional and
  fully-connected spiking networks with hidden layers, or
  locally-connected networks without hidden layers.
- Some of the arguments of the paper seem oddly fixated on the
  importance of developing spiking networks that are biologically
  plausible. For instance, on page 1: *"although convolution has shown
  great potential in solving any translation-invariant task, its use
  of weight sharing is biologically problematic"* . I agree that
  biological plausibility is important if the goal of the study is to
  learn something about biological vision, but this doesn't seem to be
  the goal of the present paper, which is instead more aligned with
  the interests of the computer vision field. In other words, several
  passages in the paper present biological plausibility as something
  desirable per se, but this is not a self-evident truth. Relatedly, the paper seems very
  quick to dismiss or ignore alternative (and very successful) methods
  for training SNNs, such as surrogate gradient-based ones (see for
  instance https://doi.org/10.1109/MSP.2019.2931595 and
  https://doi.org/10.1038/s41467-020-17236-y), because of their lack
  of biological plausibility (see for instance on page 2: *"the
  majority of works in this area use derivations of the Hebbian
  learning rule where changes in connection weights depend on the
  activities of the pre and post-synaptic neurons"* - is there a
  reference for this statement?).

## Minor points
- The introduction contains a few surprising and unsubstantiated
  statements that are likely to elicit surprise and confusion in the
  reader. For instance, *"in many learning problems, we do not have
  direct access to the explicit label of the data. Consequently, we
  may need to abandon gradient-based methods, and utilize
  reinforcement and reward-modulated learning rules"*: this seems
  completely gratuitous, and it is not clear how going gradient-free
  is a necessary implication of not having access to labels, as the
  wealth of existing gradient-based unsupervised learning methods
  shows.
- When discussing biological plausibility, I do not understand the
  insistence on the primate brain specifically. What is special about
  the primate brain that is relevant to the arguments made in this
  paper, and that does not apply, say, more generally to mammal or
  vertebrate brains?
- On page 3, please clarify in what sense *"Spike-timing-dependent
  plasticity is a type of biological Hebbian learning rule that is
  also aligned with human intuition"*. Is this supposed to imply that
  other types of biological hebbian learning rules are not "aligned
  with human intuition"? What does "aligned with human intuition"
  mean?

## Typos/clarity
- It would help clarity if the final paragraph of the discussion
  started with something like "In the future...", to signpost that
  further developments are now being discussed.
- Throughout the whole paper, please use the appropriate latex
  citation commands - I believe that in most cases you will want to
  use \citep instead of \cite or \citet. In the current state of the
  paper, many sentences are made needlessly hard to parse because the
  name of the authors of some reference is just inserted into the
  sentence without any parenthesis.
- page 1: "although convolution has shown" → "although convolutional
  networks.."
- page 2: "online on-cheap" training → "on-chip"
- page 2: "this differs with artificial neurons" → "differs from"
- page 3: "adapt the learning rule with the excitatory to
  inhibitory..." → "adapt the learning rule to"
- page 4, above eq 5: I believe "post-synaptic" neuron should be
  "pre-synaptic".
- page 4, below eq 5: "pre-syanptic"
- page 4, below eq 10: "can be considered a special CASE of R-STDP"



**Summary Of The Paper:**

This paper presents a spiking neural network (SNN) architecture for
image classification, designed to be more biologically plausible than
comparable existing architectures. In particular, it eschews
convolutional connectivity in favour of local connectivity, and uses
STDP (vanilla and reward-modulated) instead of gradient-based
learning. The paper describes in detail the proposed architecture
(providing introductory material on topics such as spiking artificial
neurons and STDP), and then compares its performance on an MNIST
benchmark to other SNN architectures. Finally, the paper presents the
performance of the architecture on a conditioning task, which is used
to highlight the capacity of the network to track a dynamically
changing reward.


**Summary Of The Review:**

This paper presents a novel combination of existing approaches. It is
in my opinion incremental in nature, but reasonably consistent in
achieving its goals. In its current state I can't recommend it for
acceptance, but I may revise my score if the confusing biological
plausibility angle is softened and if better connections are made with
related work, especially gradient based methods which at the moment
seem unjustly penalized.


---
Update after discussion: I am updating my recommendation to "marginally above the acceptance threshold" to reflect the effort made by the authors to engage at least with some of my concerns. See my message downthread for more details.

---

> ### Author Response · Authors · 2021-11-18
> **Reply to the comments raised by Reviewer b4gF**
>
> We are grateful for your comments. They gave us great insight to improve our work.
> First of all, since one of the other reviewers encountered issues running the code on colab, we have provided a new version of the code. You only need to run the “main” notebook after uploading the other files to the colab local directory and you will observe an accuracy growth similar to the blue curve in Figure 4 of the paper. As stated in the paper, BioLCNet was trained in a layerwise fashion, so we have also put the file of the pre-trained weights of the local connection, and it is automatically loaded when running the notebook. The whole training (10000 MNIST training samples) on colab GPU should take about 9 hours. Afterward, you could also run the next cell for testing on the whole MNIST test set (The time required is 256*2 since the agent does not have a learning phase).
>
> Also, a modified version of the paper is uploaded in which some of the minor issues and typos mentioned by you and the other reviewers were fixed.
>
> Clarification about a possible confusion: Equation 11 is the static rewarding mechanism, whereas equation 12 is the dynamic RPE mechanism, and they are different. Equation 12 updates the reward value to be used in the current iteration based on the reward prediction error theory. R_{i-1} is the reward value of the last step in this mechanism, and EMA is over 10 iterations. To avoid confusion, we changed r to capital R in equation 12 in the modified version.
>
> Some clarifications after reading your review:
> 1) We apologize if the paper flow conveys the feeling that it is a combination of incremental tweakings. Our main goal was to combine three bio-plausibility criteria in deep learning that have never been tried together, including spiking neurons, local connections, and reward-modulated learning. Discussing previous studies trying one or two of these criteria might have been the main cause for conveying this feeling.
>
> 2) You are completely right about the paper you mentioned [3]. The time of publication of the paper might have been the reason we missed it. Although they used only three classes of MNIST, and a small sample size of these three classes to create an RL task, it can be considered a valid case to nullify our statement. We should have cited this paper, and we removed the statement in the new version.
>
> 3) You are correct that our focus on bio-plausibility regarding backpropagation and convolutional connections ([4], [5]) should be supported by more neuroscientific statements about biological vision. In the statement "the majority of works in this area use derivations of the Hebbian learning rule where changes in connection weights depend on the activities of the pre and post-synaptic neurons", we used the word majority based on our literature review and some of the survey papers on SNNs [1], [2]. However, as you said we should have paid more attention to other methods such as surrogate gradient-based techniques. We modified the “related work” section mildly to partially address this issue.
>
> 4) “This seems completely gratuitous, and it is not clear how going gradient-free is a necessary implication of not having access to labels, as the wealth of existing gradient-based unsupervised learning methods shows.” Thank you. The sentence was rather misleading and is now modified.
>
> 5) We did not have any particular intentions insisting on the primate brain and now we modified the paper. We apologize if it has conveyed a particular intention.
>
> 6) Other Hebbian rules are also aligned with human intuition. By aligned with human intuition, we meant that the general Hebbian theory "Cells that fire together wire together." is intuitive.
> Typos and clarity issues are fixed. We also changed \citet to \citep.
>
> [1] Pfeiffer, Michael, and Thomas Pfeil. "Deep learning with spiking neurons: opportunities and challenges." Frontiers in neuroscience 12 (2018): 774.
>
> [2] Tavanaei, Amirhossein, et al. "Deep learning in spiking neural networks." Neural Networks 111 (2019): 47-63.
>
> [3] Weidel, Philipp, Renato Duarte, and Abigail Morrison. "Unsupervised learning and clustered connectivity enhance reinforcement learning in spiking neural networks." Frontiers in computational neuroscience 15 (2021): 18. doi: https://doi.org/10.3389/fncom.2021.543872
>
> [4] Bartunov, Sergey, et al. "Assessing the scalability of biologically-motivated deep learning algorithms and architectures." arXiv preprint arXiv:1807.04587 (2018).
>
> [5] Pogodin, Roman, et al. "Towards biologically plausible convolutional networks." arXiv preprint arXiv:2106.13031 (2021).

---

> > ### Comment · Reviewer_b4gF · 2021-11-22
> > **Thank you for your response**
> >
> > I wish to thank the authors for their response to my review. The authors addressed most of those concerns that could be reasonably expected to be addressable in such a short time frame. In particular, they agreed to include a more thorough overview of related literature and to tone down certain aspects that were a bit tangential. Having said this, in my opinion the work is still somewhat incremental in nature (and that's not surprising, as it would have been very hard to do anything about it in such short time).
> >
> > Overall, I appreciate the effort made by the authors to engage with the reviews they received, and I will be updating my recommendation accordingly to "marginally above the acceptance threshold". I will not, however, push for acceptance of this paper if the other reviewers do not agree with me.

---

### Official Review · Reviewer_Urvj · 2021-11-02

**Correctness:** 2
**Technical Novelty And Significance:** 2
**Empirical Novelty And Significance:** 2
**Recommendation:** 3
**Confidence:** 4

**Main Review:**

**Strengths:**
The manuscript proposed a training scheme mixed between STDP and R-STDP and proposed reward functions for training the 3-layer SNN with R-STDP. I think it is unique. However, unique does not mean good or better.

**Weaknesses:**
1. The authors proposed a dedicated 3-layer SNN network for image classification. The validation is only limited to MNIST but with worse performance than other wildly used SNN approaches. From my point of view, it is too limited and it is not convincing enough in the effectiveness of the proposed method.
2. The authors claimed, “The proposed network is the first to employ RL … for image recognition”. Even it is true, why the community should care? From my point of view, the motivation for utilizing RL for classification is not very clear, especially since the performance is worse than other commonly used SNNs.
3. It is not clear to me why the authors mixed STDP and R-STDP for training the network? I think it would make sense to show the performance with pure STDP and pure R-STDP training.
4. The last sentence of the first paragraph of the introduction, “Spiking neural networks also require fewer labeled data….”, does not make sense to me.
     * Why SNNs require fewer labeled data?
     * Fewer labeled data and operations can also be compatible with regular GPU, right?
5. I appreciate the authors tried to bring the bio-plausible concepts to the SNN domain. However, not all bio-plausible concepts revealed by biology/neuroscience research can be directly leveraged and meaningful for the SNN research as SNN neuron models themselves are just bio-inspired and they are super simple compared to the neurons in the nervous system. I do not think combining a few bio-plausible components together without clear insights could be a meaningful research project.


**Summary Of The Paper:**

The authors proposed an SNN model, dubbed BioLCNet, for image classification. The proposed network consists of three layers: 1) input layer, accepting spike trains as input;  2) locally connected (LC) hidden layer; 3) decoding fully connected (FC) output layer. The LC layer is for feature extraction and is trained first using STDP in an unsupervised manner. The decoding output FC layer is trained second with R-STDP in a semi-supervised manner. The proposed SNN is validated with the MNIST dataset to show its effectiveness.

**Summary Of The Review:**

The main concern on my end is the proposed approach lacks clear motivation why the developed SNN network and corresponding training approach are meaningful. As an SNN researcher, I did not really see the significant value. The validation is only with MNIST, and I think it is too limited.

But, my rating is not firmed at the moment. Based on the authors' feedback, if I realize I indeed miss something important, I am flexible to change my recommendation.

---

> ### Author Response · Authors · 2021-11-18
> **Reply to the comments raised by Reviewer Urvj**
>
> We appreciate your valuable comments about our paper and we find them extremely useful to improve our work.
> First of all, since one of the other reviewers encountered issues running the code on colab, we have provided a new version of the code. You only need to run the “main” notebook after uploading the other files to the colab local directory and you will observe an accuracy growth similar to the blue curve in Figure 4 of the paper. As stated in the paper, BioLCNet was trained in a layerwise fashion, so we have also put the file of the pre-trained weights of the local connection, and it is automatically loaded when running the notebook. The whole training (10000 MNIST training samples) on colab GPU should take about 9 hours. Afterwards, you could also run the next cell for testing on the whole MNIST test set (The time required is 256*2 since the agent does not have a learning phase).
>
> Also, a modified version of the paper is uploaded in which some of the minor issues and typos mentioned by you and the other reviewers were fixed.
>
> Clarification about a possible confusion: Equation 11 is the static rewarding mechanism, whereas equation 12 is the dynamic RPE mechanism, and they are different. Equation 12 updates the reward value to be used in the current iteration based on the reward prediction error theory. R_{i-1} is the reward value of the last step in this mechanism, and EMA is over 10 iterations. To avoid confusion, we changed r to capital R in equation 12 in the modified version.
>
> Clarifications after reading your review:
> 1) The reason we mixed STDP and R-STDP as learning rules to train our network is considering two phases for object recognition, feature extraction and classification. As you know STDP usually extracts more frequent features, which are seen in the inputs, and R-STDP finds more discriminant features. Therefore, we used STDP for feature extraction and R-STDP for classification.
>
> 2) “Spiking neural networks also require fewer labeled data….”, SNNs for object recognition tasks have been empirically shown to require fewer label data. For example, you can see fig.5 in [1] or fig.4 in [2]. There is a huge difference between artificial neural nets and SNNs, as in ANNs we usually have several epochs to attain reasonable performance, but in SNNs a similar performance is usually obtained just after one or even half epoch.
>
> 3) In our opinion, training SNNs and bio-plausibility are essential, because:
>       a) Trying to mimic brain networks (even in a simplified form) allows us to understand this complex network better.
>       b) More bio-plausible networks, which are more compatible with our brain, may enable us to use them in practical cases such as rehabilitation, and other BCI applications.
>      c )Non-spiking artificial neural nets do not have temporal encoding and cannot process temporal information per se, and this is a weakness for many tasks.
>      d) Spiking neural nets are much more energy-efficient and a lot faster with neuromorphic hardware.
>      e) “combining a few bio-plausible components together without clear insights could be a meaningful research project.” We apologize if the paper flow conveys this feeling. Maybe our focus on bio-plausibility regarding backpropagation and convolutional connections ([3], [4]) should be supported by more neuroscientific statements about biological vision.
>
> [1] Saeed Reza Kheradpisheh, Mohammad Ganjtabesh, Simon J Thorpe, and Timothee Masquelier.Stdp-based spiking deep convolutional neural networks for object recognition. Neural Networks, 99:56–67, 2018.
>
> [2] Daniel J Saunders, Devdhar Patel, Hananel Hazan, Hava T Siegelmann, and Robert Kozma. Locally connected spiking neural networks for unsupervised feature learning. Neural Networks, 119: 332–340, 2019.
>
> [3] Bartunov, Sergey, et al. "Assessing the scalability of biologically-motivated deep learning algorithms and architectures." arXiv preprint arXiv:1807.04587 (2018).
>
> [4] Pogodin, Roman, et al. "Towards biologically plausible convolutional networks." arXiv preprint arXiv:2106.13031 (2021).

---

> > ### Comment · Reviewer_Urvj · 2021-11-23
> > **Thanks for your feedback!**
> >
> > I appreciate the authors' efforts in making the code work easily on the reviewers' side. I also appreciate the authors' feedback related to my questions.
> >
> > First, I agree that leveraging bio-plausible concepts in improving SNN training is a meaningful and promising direction! Moreover, no need to apologize as we are discussing. It is common that reviewers may misunderstand or disagree with the authors' claims.
> >
> > "The reason we mixed STDP and R-STDP as learning rules to train our network is considering two phases for object recognition, feature extraction and classification. As you know STDP usually extracts more frequent features, which are seen in the inputs, and R-STDP finds more discriminant features. Therefore, we used STDP for feature extraction and R-STDP for classification."
> > > I would suggest showing the experimental results based on STDP only and R-STDP only. I think it is one way to show the effectiveness of the mixed training style.
> >
> > "Spiking neural networks also require fewer labeled data….", SNNs for object recognition tasks have been empirically shown to require fewer label data. For example, you can see fig.5 in [1] or fig.4 in [2]. There is a huge difference between artificial neural nets and SNNs, as in ANNs we usually have several epochs to attain reasonable performance, but in SNNs a similar performance is usually obtained just after one or even half epoch."
> > > I still cannot be convinced about this. Less number of epochs does not mean fewer labeled data. I think training one epoch requires the same amount of labeled data as training 100 epochs. Half epoch indeed means less labeled data than one epoch. However, is it very common? If only one application scenario indicates that, I do not think it is correct to claim SNN requires fewer labeled data in general.
> >
> > The most important factor influencing my recommendation is the validation, which is too limited from my perspective. I suggest validating the proposed methods based on at least one more dataset, such as the DVS128 gesture dataset.
> >
> > Therefore, I keep my original rating.

---

> > > ### Author Response · Authors · 2021-11-23
> > > **Reply to comment by Reviewer Urvj**
> > >
> > > Thank you for your feedback.
> > > Regarding your suggestion about comparing the STDP-based results with the RSTDP-based and the mixed one, we totally agree with you, thanks for suggesting, and we'll consider it in our future research. In the latter case, I think it was a poor choice of words in my comment that caused confusion. In many SNN models (for example in our case see Figure 4, or in [1], Figure 5, or in [2] Figure 4), it can be seen that the network needs a small fraction of the total dataset to achieve its best performance, and for this reason, we said "Spiking neural networks also require fewer labeled data….".
> > >
> > > Also, thank you for your comments regarding validation and suggesting a dataset. We will certainly perform experiments on more datasets.
> > >
> > > [1] Saeed Reza Kheradpisheh, Mohammad Ganjtabesh, Simon J Thorpe, and Timothee Masquelier.Stdp-based spiking deep convolutional neural networks for object recognition. Neural Networks, 99:56–67, 2018.
> > >
> > > [2] Saunders, Daniel J., et al. "Locally connected spiking neural networks for unsupervised feature learning." Neural Networks 119 (2019): 332-340.

---

### Official Review · Reviewer_BWum · 2021-11-02

**Correctness:** 3
**Technical Novelty And Significance:** 2
**Empirical Novelty And Significance:** 2
**Recommendation:** 3
**Confidence:** 4

**Main Review:**

### Major comments

**Strengths.** The paper aims at combining multiple biologically plausible features of deep networks: spikes, local connections and RL-based error signals. This is a reasonable direction, and it’s been tried by multiple papers as shown in Tab. 3.

**Weaknesses.**

However, I struggle to see the novelty in the author’s approach: spikes and local connections alone have been tried many times (Tab.3 and also [1]). Training the output layer (rather than the whole network) with an RL-based rule is somewhat new, but I find this approach unreasonable for the following reasons:
1. The last layer is usually trained with SGD + cross-entropy to assess the quality of representations built by previous layers. So the performance of R-STDP in any case would be limited by the representations it gets from earlier layers, which are arguably more important for training networks. (This paper tries to do that too with SVM, however.)
2. There’s no reason for this approach to scale beyond MNIST, as the hardest part of training is done by a simple STDP rule. Maybe some layer-wise R-STDP can be a valid approach (akin to [2]), or a backprop-like RL error [3].

As a side point, I couldn’t run the code in colab (with PyTorch 1.8 and Bindsnet installed). Running `image_classification_experiment.py` gives `record() got an unexpected keyword argument 'n_labels'`. Disabling recording makes it go away, but then there’s a shape mismatch. And if you make the running time 256 instead of 256*3 to fix it, the accuracy doesn't improve at all.

The main result of the paper -- MNIST accuracy (Tab. 3) -- is very weak. It’s pretty straightforward to achieve 95%+ test accuracy with spiking networks, local connections and unsupervised pre-training (using SGD for predictions) [1] (Tab. 2 there). Therefore, even ignoring the potential weakness of the R-STDP in the final layer and concentrating on the STDP + SVM result (87.5%), it is clear that the network does not learn useful representations. There are multiple potential reasons for that:
1. The STDP in the first layer is at fault, which would be a bit surprising given the clarity of filters in Fig.3A. As a sanity check, you can train an SVM on the hidden layer without any pre-training, and see if it improves the results.
2. The decoding scheme is ill-fitted for SVM. I’d suggest using SGD with cross-entropy like in [1] and probably many papers in Tab. 3 of your paper. If you see a large improvement, then R-STDP needs some rethinking to properly make use of the pre-trained layer.
3. Local connections make it harder. Some works in Tab. 2 of [1] successfully use LC layers, however. I would test performance with the same architecture, but using convolutions. Another thing I noticed is really large filters -- 15x15 filters for a 28x28 image are not too far from a fully connected layer.
4. When the winner-take-all in the LC layer makes a mistake by activating the wrong “digit” (and the filter weights do look like digits in Fig.3A), the readout layer can’t fix it.

Finding the root of poor performance would improve the paper, but the overall approach (hidden layer STDP + last layer R-STDP) is still unlikely to scale to harder problems and deeper networks.

**Recommendation**

Due to limited novelty and unsatisfying results, I would recommend rejecting the paper.

### Minor comments

> It is noteworthy that in many learning problems, we do not have direct access to the explicit label of the data. Consequently, we may need to abandon gradient-based methods, and utilize reinforcement and reward-modulated learning rules

Gradient-based doesn’t mean it uses labels. See VAEs, self-supervised learning, etc. that all use backprop.

> The proposed network is … the first locally-connected SNN with a hidden layer

That’s not true. See [1] and references therein.

Various problems:
1. Eq. 3 has extra underscores. It has to be dg/dt.

1. P_ij^+- in Eqs. 7-8 only need one index, j for Eq. 7 and i for Eq. 8.

1. Eq. 12 is confusing. Where does the reward come from at each trial? Is one of the r_i taken from Eq. 11?

1. Explaining the network model in Sec. 4.2 with equations would greatly improve clarity.

[1] https://www.sciencedirect.com/science/article/pii/S0893608019301741

[2] https://www.frontiersin.org/articles/10.3389/fnins.2018.00608/full

[3] https://proceedings.neurips.cc/paper/2020/hash/1abb1e1ea5f481b589da52303b091cbb-Abstract.html


**Summary Of The Paper:**

The paper discusses learning in biological networks, and proposes a model that combines spiking networks, local connections (convolutions without weight sharing) and reward-modulated spike-timing dependent plasticity. All three components are chosen to be more realistic compared to artificial neural networks. The model is tested on MNIST and on a classical conditioning task.

**Summary Of The Review:**

The paper presents a somewhat novel combination of biologically plausible features of deep nets. However, the overall approach is too simple to scale to hard tasks (STDP in hiddden layers + R-STDP in the last layer), and it performs poorly even on MNIST. The paper should be rejected.

---

> ### Author Response · Authors · 2021-11-18
> **Reply to the comments raised by Reviewer BWum**
>
> Thank you for your valuable comments and sorry for the inconvenience you encountered when running the code. The problem with the number of time steps was due to a modification we made to BindsNet monitors in our local machine for storing spikes in each iteration. The total time should be 256*3 as stated in the paper. We have attached a version of the code tested on Colab. You only need to run the “main” notebook after uploading the other files to the colab local directory and you will observe an accuracy growth similar to the blue curve in Figure 4 of the paper. As stated in the paper, BioLCNet was trained in a layerwise fashion, so we have also put the file of the pre-trained weights of the local connection, and it is automatically loaded when running the notebook. The whole training (10000 MNIST training samples) on colab GPU should take about 9 hours. Afterward, you could also run the next cell for testing on the whole MNIST test set (The time required is 256*2 since the agent does not have a learning phase).
>
> Also, a modified version of the paper is uploaded according to your comments.
>
> Clarifications after reading your review:
> 1) In [1] and [2] we can see examples combining STDP and R-STDP with good performances on various datasets, including MNIST (see [2]). They both use rank-order encoding instead of rate-based encoding. In [1], the feature extraction convolutional layer is trained with STDP, and the classifier layer with R-STDP. However, in [2] since they have a deeper architecture, for the low-level features they use STDP and for the last convolutional layer and the classification layer they use R-STDP. As an extension, we are planning on creating a deeper locally-connected network and to also try layer-wise R-STDP or at least R-STDP in the final layers as an alternative method of training.
>
> 2) In [1] they test the STDP feature-extraction and R-STDP classification combination on datasets that are more complex than MNIST. Still, as you stated, for locally-connected filters, it may be harder to achieve a performance as good as convolutional filters with this approach.
>
> 3) Our main goals for using SVM were first to perform a sanity check on the extracted features and second, error analysis. However, as you said, it is possible to get better accuracy with SVM without pre-training. We also think that modifying the classification layer architecture or reward function may improve the performance. Also, given that we have only one LC layer, using smaller filters did not achieve a good accuracy (similar to [4]).
>
> 4) “It is noteworthy that in many learning problems, we do not have direct access to the explicit label of the data. Consequently, we may need to abandon gradient-based methods, and utilize reinforcement and reward-modulated learning rules Gradient-based doesn’t mean it uses labels. See VAEs, self-supervised learning, etc. that all use backprop.”
> You are correct. This sentence was misleading and was modified in the new version.
>
> 5) “The proposed network is … the first locally-connected SNN with a hidden layer That’s not true. See [1] and references therein.”
> You are right. We had studied and cited this paper [3]. The sentence was modified.
>
> 6 ) “Eq. 3 has extra underscores. It has to be dg/dt.” Thank you for your detailed comments. Corrected.
>
> 7) “P_ij^+- in Eqs. 7-8 only need one index, j for Eq. 7 and i for Eq. 8.”
> Yes, given the models that are usually used in practice (where time constant and learning rates are the same for all synapses), we do not need two indices. Thank you. The text was modified.
>
> 8) “Eq. 12 is confusing. Where does the reward come from at each trial? Is one of the r_i taken from Eq. 11?”
> Equation 11 is the static rewarding mechanism, whereas equation 12 is the dynamic RPE mechanism, and they are different. Equation 12 updates the reward value to be used in the current iteration based on the reward prediction error theory. R_{i-1} is the reward value of the last step in this mechanism, and EMA is over 10 iterations. To avoid confusion, we changed r to capital R in equation 12 in the modified version.
>
> 9) “Explaining the network model in Sec. 4.2 with equations would greatly improve clarity.”
> Thank you. We will try to improve the clarity of our work by taking into account this suggestion.
>
> [1] Mozafari, Milad, et al. "First-spike-based visual categorization using reward-modulated STDP." IEEE transactions on neural networks and learning systems 29.12 (2018): 6178-6190.
>
> [2] Mozafari, Milad, et al. "Bio-inspired digit recognition using reward-modulated spike-timing-dependent plasticity in deep convolutional networks." Pattern recognition 94 (2019): 87-95.
>
> [3] Illing, Bernd, et al. "Biologically plausible deep learning—But how far can we go with shallow networks?." Neural Networks 118 (2019): 90-101.
>
> [4] Daniel J Saunders, et al. Locally connected spiking neural networks for unsupervised feature learning. Neural Networks, 119: 332–340, 2019.

---

> > ### Comment · Reviewer_BWum · 2021-11-22
> > **Minor comments addressed, by the major ones are stiill there; levaing the same score (3)**
> >
> > Thank you for the response!
> >
> > Most of my minor comments have been addressed, but not the major ones (the first numbered list in the original review). To sum up those complaints, the proposed method doesn't really tell us how to train deep networks, just the output layer. But training the output layer is mostly a question of how much supervision signal you get, and is somewhat orthogonal to spiking/local connections/other biologically realistic properties. I think adapting R-STDP to all layers might be interesting (although I suspect it won't scale well), but the current results are not, unfortunately.
> >
> > **I therefore still recommend rejection, and leave the same score (3).**
> >
> > Also, re: eq.12. I still don't understand where the actual rewards comes into play. With the new notation, shouldn't it be $R_i = R_{i-1} - \eta (r_{i-1} - EMA)$?

---

> > > ### Author Response · Authors · 2021-11-22
> > > **Reply to comments by Reviewer BWum**
> > >
> > > Thank you for your response. We hope you understand that in this short time frame we could not perform many additional experiments and parameter tuning, or alter the submission radically (such as adding more layers, using R-STDP for feature extraction, additional SVM analysis, or checking our model scalability with a more complex dataset). However, we are certain that your comments will be valuable in our further works and experiments.
> > >
> > > Again, sorry about equation 12. The final text that we just uploaded now, reads:
> > > We formulate our dynamic RPE mechanism as below,
> > >
> > > $R_i = R_{i-1} - \eta_{rpe}(r_{i}-\mathrm{EMA}_R)$
> > >
> > > where $R_i$ is the scalar R-STDP modulation signal used during the whole learning period ($T_{learn}$) of the $i^{th}$ training sample, $r_{i}$ is the reward signal received based on the prediction of the network, and $\mathrm{EMA}_R$ is the exponential moving average of the modulation signals with a smoothing factor $\alpha$.
> > >
> > > Indeed, the actual reward signal is in fact $r_{i}$ inside the parentheses, and based on the prediction could be 1 or -1. It might have been better if we focused and elaborated more on rewarding in general. To our knowledge, the use of R-STDP up to now has been limited to only static reward signals. At least in our case, the use of a dynamic rewarding mechanism (eq12) instead of a static one (eq11) resulted in a performance jump (compare Fig 4 green and blue plots). We also devised and experimented with many other dynamic rewarding mechanisms that we built on top of BindsNet (you can see "reward.py" in supplementary materials), such as rewards proportional to the number of spikes in the classifier layer neuron groups. We hope that in the future, we see more RL-based SNNs with novel and efficient rewarding schemes in the literature.

---

### Decision · Program_Chairs · 2022-01-20

**Decision:**

Reject

**Comment:**

This paper presents a locally connected spiking neural network model trained to do classification of MNIST using spike-timing-dependent plasticity (STDP) and reward-modulated STDP. The authors show that this model can learn to classify MNIST images (though not at a very high accuracy) and that it can engage in classical conditioning. The reviews were initially all in the reject range. The common theme in the reviews was concerns about the weak and limited nature of the results. After a good amount of author response and reviewer replies to the authors, one reviewer increased their score to a borderline accept, but the other reviewers did not change their scores, producing scores of 3,3, and 6. Given these scores, and the reviewers' remaining concerns, a reject decision was reached.